# Focal Photocoagulation as an Adjunctive Therapy to Reduce the Burden of Intravitreal Injections in Macula Edema Patients, the LyoMAC2 Study

**DOI:** 10.3390/pharmaceutics15020308

**Published:** 2023-01-17

**Authors:** Lucas Séjournet, Laurent Kodjikian, Sandra Elbany, Benoit Allignet, Emilie Agard, Mayeul Chaperon, Jérémy Billant, Philippe Denis, Thibaud Mathis, Carole Burillon, Corinne Dot

**Affiliations:** 1Department of Ophthalmology, Desgenettes Military Hospital, 69003 Lyon, France; 2Department of Ophthalmology, Hôpital de la Croix-Rousse, Hospices Civils de Lyon, 69004 Lyon, France; 3Centre de Recherche Clinique, Hôpital de la Croix-Rousse, Hospices Civils de Lyon, 69004 Lyon, France; 4Department of Ophthalmology, Hôpital Edouard Herriot, Hospices Civils de Lyon, 69003 Lyon, France; 5Department of Radiation Oncology, Centre Léon Bérard, 28 rue Laennec, 69673 Lyon, France; 6CREATIS, CNRS UMR 5220, Inserm U1206, INSA-Lyon, Université Jean Monnet Saint-Étienne, Université Claude Bernard Lyon 1, 69621 Villeurbanne, France; 7French Military Academy of Val de Grace, 75005 Paris, France

**Keywords:** macular edema, capillary macroaneurysms, telangiectatic capillaries, diabetic macular edema, retinal vein occlusion, focal laser photocoagulation

## Abstract

Aim: To assess the efficacy of focal photocoagulation of capillary macroaneurysms (CMA) to reduce the burden of intravitreal injections (IVI) in patients with macular edema (ME). Materials and Methods: Retrospective multicenter study in patients with diabetic ME or ME secondary to retinal vein occlusion (ME-RVO). CMA associated with ME were selectively photocoagulated. Patients were followed for one year after photocoagulation. Results: 93 eyes of 76 patients were included in this study. At 6 months after the laser (n = 93), there was a significant decrease in mean macular thickness (from 354 µm to 314 µm, *p* < 0.001) and in mean IVI number (from 2.52 to 1.52 at 6 months, *p* < 0.001). The mean BCVA remained stable (0.32 and 0.31 logMAR at baseline and 6 months, *p* = 0.95). At 12 months (n = 81/93), there was a significant decrease in mean macular thickness (from 354 µm to 314 µm, *p* < 0.001) and in mean IVI number (from 4.44 to 2.95 at 12 months, *p* < 0.001), while the mean BCVA remained stable (0.32 and 0.30 logMAR at baseline and 12 months, *p* = 0.16). Conclusion: Focal laser photocoagulation of CMA seems to be effective and safe for reducing the burden of IVI in patients with ME. Their screening during the follow-up should be considered closely.

## 1. Introduction

Diabetic retinopathy and retinal vein occlusion (RVO) are the leading causes of long-lasting macular edema (ME), which can lead to an irreversible loss of vision in the absence of treatment [1,2].

Focal vascular abnormalities such as capillary microaneurysms may develop in these retinal vascular diseases [3,4]. The diameter of these microaneurysms is usually less than 90 µm; however, the diameter of some aneurysms can reach up to several hundred microns [5,6]. These large abnormalities, referred to as capillary macroaneurysms (CMA) [6,7] or telangiectatic capillaries [8] in the medical literature, may be associated with chronic refractory ME and hard exudates [5]. The prevalence of CMA in diabetic ME (DME) and RVO is about 60%, and indocyanine green angiography (ICGA) and optical coherence tomography (OCT) have been previously shown to effectively detect CMA [9,10].

Treatment of ME is mainly based on intravitreal injections (IVIs) of anti-vascular endothelial growth factor (VEGF) or corticosteroids. Because of its chronicity, IVIs and physical examinations are needed in the long term, leading to a significant therapeutic burden. In recent years, the main objective has been to increase the interval between IVIs through the use of new injection protocols such as the treat-and-extend regimen [11,12]. A potential alternative could be to photocoagulate CMA.

Several publications have shown that focal laser photocoagulation of CMA effectively improves the visual acuity and reduces the macular thickness [6,13,14,15]. These studies are mostly retrospective, non-controlled, non-blinded, and monocentric. However, in these previous studies, the sample size was small, and no patients previously treated with IVIs were included. None of these studies has assessed the long-term efficacy of photocoagulation or the need for IVIs after focal laser photocoagulation.

The aim of this retrospective study was to assess the relevance of focal laser photocoagulation of CMA in both DME and ME-RVO to reduce the burden of intravitreal injections in patients with long-lasting macular edema (ME) by determining the number of IVIs received after laser treatment. The secondary outcomes were the best-corrected visual acuity (BVCA), the focal retinal thickness, and the macular thickness at 6 and 12 months after focal laser photocoagulation.

## 2. Materials and Methods

### 2.1. Patient Selection

A multicenter, retrospective, non-randomized, and non-comparative study was conducted in adult patients with DME or ME-RVO treated with focal laser photocoagulation for CMA between November 2019 and June 2022 in the ophthalmology departments of Desgenettes Military Hospital, Croix Rousse, and Edouard Herriot University Hospital, Lyon (France). This study was conducted in accordance with the principles outlined in the Declaration of Helsinki. All patients received oral and written information and gave their consent before being treated for any laser treatment. The Ethics Committee of the French Society of Ophthalmology approved the study conduct (IRB 00008855 Société Française d’Ophtalmologie IRB 1).

### 2.2. Data Collection

All patients treated with focal laser photocoagulation were included in this study, only if they had been treated with IVIs at least 12 months prior to the laser treatment. The medical history, the number and type of IVIs received, and prior ophthalmological findings were recorded retrospectively. The dataset was collected using anonymized Excel files before statistical analysis.

The systematic screening of CMA has been standardized in our departments since 2019. In all patients, an ophthalmological examination and retinal imaging, including fluorescein angiography (FA), ICGA, and OCT (spectral domain [SD]-OCT, Heidelberg Engineering, Heidelberg, Germany), were performed. CMA were identified on the late-phase ICGA and on the OCT scans (Figure 1). All lesions related to ME located more than 750 µm from the fovea were treated. Patients with CMA in the central 1500-µm macular area (<750 µm) were also included if other lesions were accessible for focal photocoagulation. The CMA diameter was measured on the OCT B-scan (Spectralis HRA, “pole post” pattern, scan length of 8.3 mm). The mean central macular thickness, the mean focal edema thickness (i.e., the thickness of the retina around the CMA), and the mean distance between the CMA and the fovea were also recorded.

### 2.3. Outcome Measures

Laser photocoagulation was performed with a Centralis® contact lens (Volk) using a 532-nm laser with the following parameters: spot size of 50 µm; duration of 0.2 s. The power was increased from 100 mW until whitening of the macroaneurysms was observed. Three consecutive impacts were made. In our daily practice, OCT is not performed after laser therapy.

IVIs were administered according to a treat-and-extend protocol with a 2-week adjustment period. Injection intervals were decreased when the retinal fluid was stable or increased on the OCT B-scan or in cases of worsening of the BCVA. Injection intervals were increased in the absence of retinal fluid on the OCT B-scan. Treatment intervals were clinically assessed by an ophthalmologist at each visit. Aflibercept, ranibizumab, and dexamethasone implant (DEXi) were used for both DME and ME-RVO. A treatment switch was possible in the absence of response after three consecutive anti-VEGF IVIs. IVIs could be discontinued when a dry macula persisted 4 months after the last injection of aflibercept and ranibizumab and 6 months after the last DEXi injection.

Follow-up examinations were performed 3, 6, and 12 months after photocoagulation. Opacification of the CMA lumen was assessed by comparing the pre- and post-laser OCT scans using the eye tracking tool. Patients could be retreated in the absence of signs of photothrombosis and when intraretinal fluid persisted 3 months after laser treatment.

### 2.4. Statistics

Data are presented as a mean ± standard deviation (SD) or a mean (range) and as a count (percentage) for quantitative and qualitative variables, respectively. A paired-sample Wilcoxon rank-sum test or a t test was used for comparisons of continuous variables. All tests were two-sided, and a *p*-value < 0.05 was considered significant. Statistical analyses were performed using R software version 4.1.2 (R Foundation for Statistical Computing, Vienna, Austria).

## 3. Results

### 3.1. Patients’ Characteristics

Ninety-three eyes of 76 patients were included in this study. Patients’ characteristics are shown in Table 1. The mean age was 70.8 (range: 31–90), and 44 patients (56.6%) were men. Regarding the distribution, 62 cases were DME and 31 cases were ME-RVO.

The mean ME duration before laser treatment was 23.9 months (range: 12–48). All patients received IVIs for at least 12 months before inclusion. Patients received a mean number of 4.43 ± 2.44 IVIs and 2.52 ± 1.44 IVIs in the 12 and 6 months prior to photocoagulation, respectively. Patients were treated with anti-VEGF (39/93, 39.8%), DEXi (39/93, 41.9%), or both (17/93, 18.3%) in the 12 months prior to photocoagulation. At baseline, the mean BCVA was 0.32 ± 0.15 logMAR, the mean central macular thickness was 354 ± 114 µm, and the mean focal retinal thickness was 462 ± 92.7 µm. The mean number of CMA per eye was 2.6 (range: 1–7). The mean diameter of CMA was 171 ± 52 µm. The mean closest distance between the lesions and the fovea was 1690 µm (range: 750–3100 µm). At baseline, hard exudates and circinate exudates were detected in 71/93 and 14/93 eyes, respectively.

### 3.2. Primary and Secondary Outcomes

Six months after photocoagulation, there was a significant decrease in the mean number of IVIs received (from 2.52 IVIs in the 6 months prior to photocoagulation to 1.52 IVIs 6 months after photocoagulation, *p* < 0.001), in the mean foveal thickness (from 354 µm at baseline to 314 µm at 6 months, *p* < 0.001), and in the mean focal thickness (from 462 µm at baseline to 399 µm at 6 months, *p* < 0.001). The mean BVCA was not significantly improved six months after photocoagulation compared to the baseline (0.32 and 0.31 logMAR, respectively; *p* = 0.95). The same results were observed 12 months after photocoagulation for the mean number of IVIs (from 4.44 IVIs in the 12 months before photocoagulation to 2.95 IVIs 12 months after photocoagulation, *p* < 0.001), the mean foveal thickness (from 354 µm at baseline to 299 µm at 12 months, *p* < 0.001), and the mean focal thickness (from 462 µm at baseline to 399 µm at 12 months, *p* < 0.001). The mean BVCA was not significantly improved 12 months after photocoagulation compared to baseline (0.32 and 0.30 logMAR, respectively; *p* = 0.16) (Figure 2 and Figure 3).

Differences in central retinal thickness and focal thickness are statistically significant (*p* < 0.001). The BCVA was not significantly improved (*p* > 0.05).

CMA were still detected in 15/93 eyes at 6 months because they were too close to fovea (4/15), or they could not be targeted due to poor patient fixation, cataract, or vitreous hemorrhage (11/15). At 6 months, this subgroup of patients with persistent CMA (CMA+) received a mean number of 2.11 IVIs compared to 1.37 IVI in patients with occluded CMA (CMA-) (*p* = 0.02), had a poorer BVCA (0.37 versus 0.29 logMAR in the CMA+ and CMA- subgroups, *p* < 0.01), a higher foveal thickness (354 versus 303 µm in the CMA+ and CMA- subgroups, *p* = 0.01), and a higher focal thickness (469 versus 382 µm in the CMA+ and CMA- subgroups, *p* < 0.01). The same results were observed at 12 months with a mean number of 4.1 IVIs in the CMA+ subgroup versus 2.6 IVIs in the CMA- subgroup (*p* = 0.04), a higher foveal thickness (323 versus 291 µm in the CMA+ and CMA- subgroups, *p* < 0.01), a poorer BVCA (0.39 versus 0.27 logMAR in the CMA+ and CMA- subgroups, *p* < 0.01), and a higher focal thickness (449 versus 385 µm in the CMA+ and CMA- subgroups, *p* < 0.01).

Hard exudates had disappeared in 26/71 (37%) eyes at 6 months and in 34/66 (50%) eyes at 12 months. Overall, IVIs were fully discontinued in 23/93 eyes in 21/76 (27.6%) patients. The results of the primary and secondary outcomes are shown in Table 2.

At 12 months, treatment was switched in 5/93 eyes from anti-VEGF to DEXi (n = 4) and from DEXi to aflibercept (n = 1), and three patients received an acetonide fluocinolone implant in addition to DEXi. In 30/93 (32%) eyes of 28 patients, laser photocoagulation was repeated because OCT showed CMA with persistent retinal fluid. In 5/30 eyes of 5 patients, laser photocoagulation was repeated a third time. No complications from photocoagulation were reported.

### 3.3. Functional and Anatomical Outcomes According to the Type of IVI Agent

After 6 months, patients treated with anti-VEGF showed a significant decrease in the mean number of IVIs (from 3.62 IVIs 6 months before photocoagulation to 1.9 IVIs at 6 months, *p* < 0.001), in the mean central foveal thickness (from 301 µm at baseline to 283 µm at 6 months, *p* = 0.04) and in the mean focal thickness (from 416 µm at baseline to 376 µm at 6 months, *p* < 0.001) compared to baseline. There was no significant difference in BVCA six months after photocoagulation (0.22 and 0.25 logMAR at baseline and at 6 months; *p* = 0.35). After 12 months, patients treated with anti-VEGF showed a significant decrease in the mean number of IVIs (from 6.19 IVIs 12 months before photocoagulation to 3.70 IVIs at 12 months, *p* < 0.001), in the mean central foveal thickness (from 301 µm at baseline to 237 µm at 12 months, *p* < 0.01), and in the mean focal thickness (from 416 µm at baseline to 358 µm at 12 months, *p* < 0.01). The BVCA was not significantly improved 12 months after photocoagulation compared to baseline (0.22 and 0.24 logMAR at baseline and 6 months; *p* = 0.80).

After 6 months, patients treated with DEXi showed a significant decrease in the mean number of IVIs (from 1.69 IVIs 6 months before photocoagulation to 1.26 IVIs at 6 months, *p* = 0.02), in the mean central thickness (from 403 µm at baseline to 336 µm at 6 months, *p* < 0.01), and in the mean focal thickness (from 492 µm at baseline to 427 µm at 12 months, *p* < 0.01). There was no significant difference in BVCA six months after photocoagulation (0.47 and 0.43 logMAR at baseline and 6 months, *p* = 0.08). After 12 months, there was a significant decrease in the mean number of IVIs (from 3.05 IVIs 12 months before photocoagulation to 2.42 IVIs at 12 months, *p* = 0.02), in the mean central thickness (from 403 µm at baseline to 317 µm at 12 months, *p* < 0.01), and in the mean focal thickness (from 492 µm at baseline to 425 µm at 12 months, *p* < 0.01). The BVCA was not significantly improved compared to baseline (0.47 and 0.39 logMAR at baseline and 12 months, *p* = 0.11). These results are shown in Table 3.

### 3.4. Functional and Anatomical Outcomes According to the Size of CMA

CMA larger than 150 µm were found in 60/93 eyes. A total of 6 months after photocoagulation, there was a significant decrease in the mean number of IVIs (from 2.30 IVIs 6 months before photocoagulation to 1.40 at 6 months, *p* < 0.001), in the mean central foveal thickness (from 373 µm at baseline to 317 µm at 6 months, *p* < 0.001), and in the mean focal thickness (from 474 µm at baseline to 401 µm at 6 months, *p* < 0.001). The BVCA was not significantly improved 6 months after photocoagulation compared to baseline (0.36 and 0.34 logMAR at baseline and at 6 months; *p* = 0.97). Similar results were obtained 12 months after photocoagulation for the mean number of IVIs (from 4.05 IVIs 12 months before photocoagulation to 2.84 IVIs at 12 months, *p* < 0.001), the mean foveal thickness (from 373 µm at baseline to 309 µm at 12 months, *p* < 0.001), and the mean focal thickness (from 474 µm at baseline to 402 µm at 12 months, *p* < 0.001). The BVCA was not significantly improved 12 months after photocoagulation compared to baseline (0.36 and 0.33 logMAR at baseline and at 12 months; *p* = 0.16).

CMA smaller than 150 µm were found in 33/93 eyes. A total of 6 months after photocoagulation, there was a significant decrease in the mean number of IVIs (from 2.91 IVIs 6 months before photocoagulation to 1.73 IVIs at 6 months, *p* < 0.001) and in the mean focal thickness (from 441 µm at baseline to 397 µm at 6 months, *p* < 0.001). The BVCA and the mean central foveal thickness were not significantly improved six months after photocoagulation compared to baseline (0.26 and 0.26 logMAR at baseline and at 6 months, *p* = 0.95, and 319 and 306 µm at baseline and at 6 months, *p* = 0.34, respectively). Similar results were obtained 12 months after photocoagulation for the mean number of IVIs (from 5.15 IVIs 12 months before photocoagulation to 3.13 IVIs at 12 months, *p* < 0.01), the mean foveal thickness (from 319 µm at baseline to 280 µm at 12 months, *p* < 0.01), and the mean focal thickness (from 441 µm at baseline to 395 µm at 12 months, *p* < 0.01). The BVCA was not significantly improved 12 months after photocoagulation compared to baseline (0.26 and 0.25 logMAR at baseline and at 12 months; *p* = 0.60). These results are shown in Table 4.

### 3.5. Functional and Anatomical Outcomes According to the Cause of ME

In DME patients, after 6 months, there was a significant decrease in the mean number of IVIs (from 2.52 IVIs 6 months before photocoagulation to 1.53 IVIs at 6 months, *p* < 0.01), in the mean central foveal thickness (from 360 µm at baseline to 319 µm at 6 months; *p* < 0.01), and in the mean focal thickness (from 476 µm at baseline to 415 µm at 6 months, *p* < 0.01). There was no significant difference in BVCA six months after photocoagulation (0.32 and 0.30 logMAR at baseline and at 6 months; *p* = 0.73). After 12 months, DME patients showed a significant decrease in the mean number of IVIs (from 4.37 IVIs 12 months before photocoagulation to 3.15 IVIs at 6 months, *p* < 0.01), in the mean central foveal thickness (from 360 µm at baseline to 299 µm at 12 months, *p* < 0.01), and in the mean focal thickness (from 476 µm at baseline to 413 µm at 12 months, *p* < 0.01). The BVCA was not significantly improved 12 months after photocoagulation compared to baseline (0.32 and 0.28 logMAR at baseline and at 12 months, *p* = 0.06).

In ME-RVO patients, after 6 months, there was a significant decrease in the mean number of IVIs (from 2.51 IVIs 6 months before photocoagulation to 1.48 IVIs at 6 months, *p* < 0.01), in mean central thickness (from 342 µm at baseline to 302 µm at 6 months, *p* = 0.02), and in mean focal thickness (from 436 µm at baseline to 371 µm at 6 months, *p* < 0.01). There was no significant difference in BVCA six months after photocoagulation (0.33 and 0.33 logMAR at baseline and at 6 months, *p* = 0.70). After 12 months, there was a significant decrease in the mean number of IVIs (from 4.58 IVIs 12 months before photocoagulation to 2.59 IVIs at 12 months, *p* < 0.01), in mean central thickness (from 342 µm at baseline to 298 µm at 12 months, *p* < 0.01), and in mean focal thickness (from 436 µm at baseline to 373 µm at 12 months, *p* < 0.01). The BVCA was not significantly improved compared to baseline (0.33 and 0.33 logMAR at baseline and at 12 months, *p* = 0.68). These results are shown in Table 5.

### 3.6. Functional and Anatomical Outcomes According to the Distance between CMA and the Fovea

The whole cohort of patients was divided into tertiles based on the closest distance between the CMA and the fovea. Patients whose closest CMA was located less than 750 µm from the fovea were excluded. The first tertile (T1) included 30 eyes with the closest distance between CMA and the fovea ranging between 750 and 1400 µm; the second tertile (T2) included 31 eyes with the closest distance between CMA and the fovea ranging between 1400 and 1900 µm; and the third tertile (T3) included 28 eyes with the closest distance between CMA and the fovea ranging between 1900 and 3100 µm.

Patients in all tertiles showed a significant decrease in the mean number of IVIs from baseline, regardless of the timepoint analyzed (at 6 and 12 months). At 6 months, the mean number of IVIs decreased from 2.17 to 1.67 in T1 (*p* = 0.03), from 2.55 to 1.51 in T2 (*p* < 0.01), and from 2.93 to 1.35 in T3 (*p* < 0.01) compared to 6 months before photocoagulation. At 12 months, the mean number of IVIs decreased from 4.17 to 3.20 in T1 (*p* < 0.01), from 4.61 to 2.80 in T2 (*p* < 0.01), and from 4.64 to 2.80 in T3 (*p* < 0.01) compared to 12 months before photocoagulation. Moreover, the number of IVIs decreased when the distance to the fovea increased, with a mild but significant correlation between the distance to the fovea and the number of IVIs at 6 months (31.3%; *p* = 0.003). However, there was no significant correlation between the distance to the fovea and the number of IVIs at 12 months (*p* = 0.42).

## 4. Discussion

In our non-comparative retrospective study, we reached our primary endpoint, showing that focal laser photocoagulation of CMA effectively reduced the number of IVIs in patients with chronic DME or ME-RVO. To our knowledge, this was the largest real-life study conducted on this topic.

In chronic ME, the grid laser has been historically performed with an overall modest effect on vision [16,17,18]. In our study, targeted, selective laser photocoagulation appeared to be a more appropriate approach. However, in current guidelines, CMA treatment is not yet explicitly mentioned [19,20]. Moreover, our subgroup analysis according to the cause of ME showed that targeting CMA was effective in both DME and ME-RVO patients.

In the literature, there is no consensus on the definition of CMA. Some authors have defined CMA as any microvascular lesion with late focal hyperfluorescence on ICGA [5] or use a diameter threshold of 150 µm [13,21]. A retrospective study conducted in 2013 found a lower efficacy of laser photocoagulation on lesions <150 µm [22]. Nevertheless, we chose the smaller size of 100 µm to treat all the lesions eligible for laser therapy, as explained in a previous study recently published by our team [9]. Indeed, it may be challenging to see and effectively target CMA <100 µm when a traditional laser is used, compared to a navigated retinal laser. We showed consistent results between patients with CMA >150 µm and <150 µm in a subgroup analysis.

It may be challenging to locate CMA for laser treatment. Only lesions located at more than 750 µm from the fovea were treated in our study to reduce the risk of lesion scar evolution and foveal damage. No complication was reported in our study after focal coagulation. Navigated retinal lasers, such as the Navilas® system, could allow treating lesions closer to the fovea if needed [21]. The fact that patients with lesions closer than 750 µm were also included in this study when another CMA was eligible for treatment could explain the poorer occlusion rate compared to other studies [13,21]. As expected, patients with persistently active CMA at six months received a higher number of IVIs than patients with fully treated CMA and had poorer BVCA and foveal and focal thicknesses.

We also showed that focal photocoagulation decreased the mean number of IVIs in patients treated with anti-VEGF and DEXi. Patients treated with DEXi had a more severe ME, as shown by their baseline BVCA, mean foveal thickness, and mean focal thickness. Additionally, it should be noted that the number of IVIs was reduced by almost 50% 12 months after laser therapy in patients treated with anti-VEGFs without any complications, whereas this reduction was less marked in patients treated with DEX-i (25%). This could be explained by the longer release of this implant (4 months) and the severity of the ME. Nevertheless, the gain in DEX injections was about 1 IVI at 12 months. In addition, IVIs could be discontinued in 25% of eyes, suggesting that, in some cases, ME was mainly due to CMA and that its appropriate diagnosis and treatment could be very relevant to decreasing or even discontinuing IVIs. On the other hand, some patients had persistently active CMA despite several focal photocoagulation sessions. This could be explained by the location of some CMA that were too close to the fovea to be targeted or by the difficulty of performing laser treatment due to a poor fixation of patients, cataract, or vitreous hemorrhage. This could have contributed to increasing the number of IVIs in these patients and decreasing the overall efficacy of treatment.

We used a 532-nm laser with standard parameters as previously described [6]. Other types of lasers have shown their efficacy in microaneurysm and DME treatment, such as yellow lasers and subthreshold micropulse lasers [23,24]. To date, no study has assessed these lasers for CMA treatment in ME-RVO and DME, despite their potential interest. Indeed, these types of lasers are not yet available in all departments.

Infrared reflectance image-guided focal laser photocoagulation of CMA has recently been shown to be effective in persistent DME [14,15]. We have highlighted the good detection of CMA on en face OCT in the LyoMAC1 study [12]. This could be, in some cases, a new multimodal way, including OCT-angiography, to detect CMA in DME and ME-RVO before planning laser treatment.

As expected, the BVCA was not improved after focal photocoagulation, despite a trend in DME patients (*p* = 0.06). However, we showed a decrease in mean macular thickness after CMA treatment. A focal laser treatment could be considered an adjuvant treatment. We could assume that combining focal laser photocoagulation with IVIs could improve the long-term functional outcomes since it is known that fluctuations are deleterious in DME [25].

IVIs are an effective but expensive treatment for DME and ME-RVO [26,27]. The cost-effectiveness of the systematic detection and laser focal photocoagulation of CMA was not assessed in this study but should raise interest because the number of IVIs was significantly reduced in our patients after one year. In addition, its effect on quality of life and decrease in therapeutic burden will also need to be assessed.

This study has some limitations. Our patients were likely to have more severe retinal disorders at baseline because they were followed at three tertiary centers. Three different university centers were involved, all in the same city, reflecting a local approach and not reflecting the diversity of medical practice worldwide. Furthermore, this was a non-blinded and non-comparative study. This can induce some measurement bias, as investigators may have been influenced by the fact that the laser was conducted and could have been less stringent about edema recurrence. Patients were under their own control during the year after laser treatment. The reduction of injections with time is well known; nevertheless, our patients had chronic and old macular edema with repeated injections for at least 2 years. Our study was also limited by the small size of our cohort, even though this is the largest series described to date. As focal photocoagulation was not performed by a single investigator, this could explain some differences between centers, but this reflects our real-life experience. The retrospective nature of this study could also have led to missing data or missing confounding factors. As several investigators were involved in this study, the treatment decision could have been partially subjective, despite being confirmed by a senior retinal specialist. Treatments could be switched, stopped, or intensified at the discretion of physicians, which could reflect to some degree the doctor’s preference. However, common guidelines for recurrences were followed in all centers to limit this bias. Moreover, we did not find a strong correlation between distance to the fovea and outcome measurements. Although it seems intuitive to think that the closer the CMA is to the fovea, the better the laser treatment would work, our results did not support this. This could be explained by the small size of sub-groups. All patients achieved our primary outcome at 6 months, but some patients were lost to follow-up before the 12-month visit. The number of patients lost to follow-up was low, and its impact on our results should be low. Switching to DEXi or a fluocinolone acetonide implant could have artificially reduced the number of IVIs over 6 and 12 months, but only a few patients underwent the switch, and it should therefore have no significant impact on our results.

The strengths of our study are the systematic use of ICGA for the diagnosis of CMA and the use of multimodal imaging for laser decisions. The same laser wavelength and protocol were used to treat. Patients were under their own control, excluding the influence of other comorbidities common in diabetes. The one-year follow-up after photocoagulation allowed us to identify a reduction in treatment burden. Additionally, our study was a real-life study reflecting what we are supposed to face in our daily practice.

Our results could help better define focal photocoagulation procedures in order to improve their efficacy. Systematic screening and treatment of CMA in chronic or resistant ME could be of interest to reduce the therapeutic burden for patients and the medical cost for society.

## 5. Conclusions

Focal laser photocoagulation of CMA in patients with DME and ME-RVO appears to be relevant to extending the interval between IVIs and reducing the burden of intravitreal injections in patients with long-lasting ME. It should be noted that the number of IVIs was reduced by almost 50%. A total of 12 months after laser therapy, in patients treated with anti-VEGFs and IVIs, could be totally discontinued in 25% of eyes. This approach could be beneficial for patients, especially in terms of quality of life and medico-economic aspects.

As our study is non-randomized, non-blinded, and retrospective, further larger prospective controlled studies are needed to confirm our results.

## Figures and Tables

**Figure 1 pharmaceutics-15-00308-f001:**
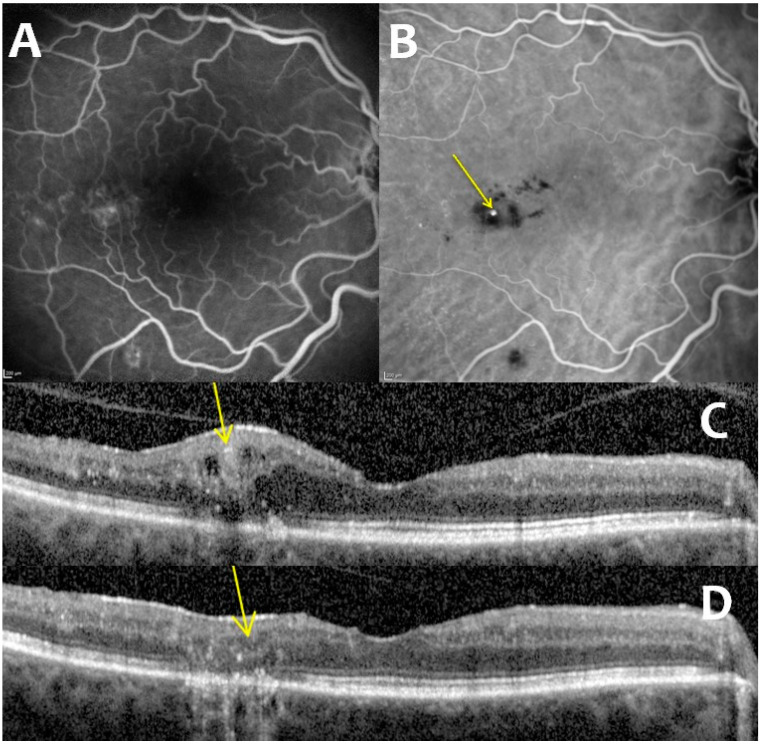
Multimodal imaging of capillary macroaneurysms: fluorescein angiography (**A**), indocyanine green angiography (**B**), and comparison of pre-laser (**C**) and post-laser (**D**) optical coherence tomography showing the disappearance of macroaneurysms (pointed out with yellow arrows).

**Figure 2 pharmaceutics-15-00308-f002:**
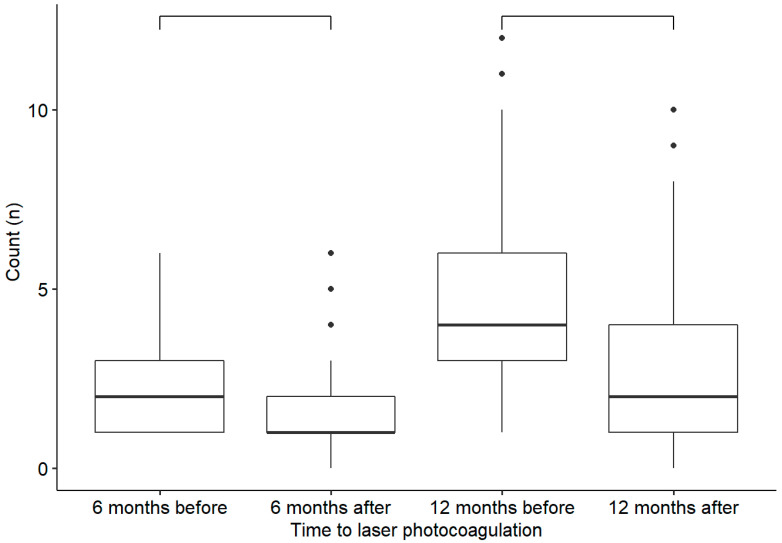
Comparison of the number of intravitreal injections received before and after laser treatment. The differences are statistically significant (*p* < 0.001).

**Figure 3 pharmaceutics-15-00308-f003:**
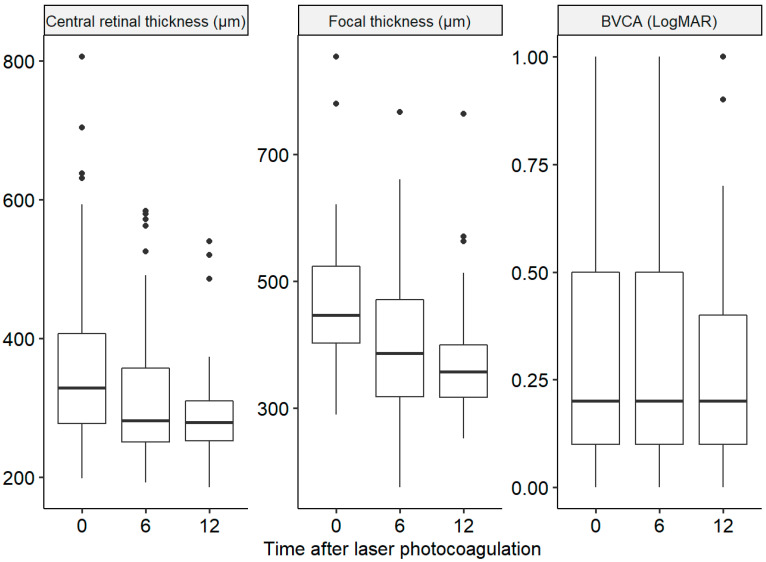
Comparison of the central retinal thickness, focal thickness, and best-corrected visual acuity (BCVA) measured at baseline and 6 and 12 months after laser treatment.

**Table 1 pharmaceutics-15-00308-t001:** Baseline clinical characteristics of the patients.

Variables	n (%) or Mean ± SD
Female gender	33 (43.4%)
Age, years	70.8 ± 11.2
Cause of macular edema	
Diabetes	62 (66.6%)
RVO	9 (9.7%)
BRVO	22 (23.7%)
Duration of macular edema	23.9 ± 11
Capillary macroaneurysms	
Number (mean)	1.97 ± 1.11
Largest size (mean), µm	172 ± 52
Distance to the fovea (mean), µm	1690 ± 664
Intravitreal injections	
Number in the last 6 months (mean)	2.52 ± 1.43
Number in the last 12 months (mean)	4.44 ± 2.44
Anti-VEGF agent	37 (39.8%)
Dexamethasone implant	39 (41.9%)
Both	17 (18.3%)
SD-OCT findings	
Foveal thickness (mean), µm	354 (114)
Focal edema thickness (mean), µm	462 (92.7)

Abbreviations: SD-OCT—Spectral-Domain Optical Coherence Tomography; DME—Diabetic Macular Edema; RVO—Retinal Vein Occlusion; BRVO—Branch Retinal Vein Occlusion; VEGF—Vascular Endothelial Growth Factor; and SD—standard deviation.

**Table 2 pharmaceutics-15-00308-t002:** Primary and secondary outcomes 6 and 12 months after photocoagulation.

Primary Outcome	6 Months before Photocoagulation	6 Months after Photocoagulation	*p*	12 Months before Photocoagulation	12 Months after Photocoagulation	*p*
Mean number of injections	2.52	1.52	<0.001	4.44	2.95	<0.001
**Secondary outcomes**	**Baseline**	**6 Months after Photocoagulation**	** *p* **	**12 Months after Photocoagulation**		*p*
Mean best-corrected visual acuity (LogMAR)	0.32	0.31	0.95	0.30		0.16
Mean foveal thickness (µm)	354	314	<0.001	299		<0.001
Mean focal thickness (µm)	462	400	<0.001	399		<0.001

**Table 3 pharmaceutics-15-00308-t003:** Subgroup analysis of the primary and secondary outcomes according to the type of intravitreal injections received (BCVA—Best-Corrected Visual Acuity; VEGF—Vascular Endothelial Growth Factor).

Anti-VEGF						
	6 months before photocoagulation	6 months after photocoagulation	*p*	12 months before photocoagulation	12 months after photocoagulation	*p*
Mean number of injections	3.62	1.90	<0.001	6.19	3.70	<0.001
	Baseline	6 months after photocoagulation	*p*	12 months after photocoagulation		*p*
Mean BCVA (LogMAR)	0.22	0.25	0.35	0.24		0.80
Mean foveal thickness (µm)	301	283	0.04	273		<0.01
Mean focal thickness (µm)	416	358	<0.01	358		<0.01
**Dexamethasone implant**						
	6 months before photocoagulation	6 months after photocoagulation	*p*	12 months before photocoagulation	12 months after photocoagulation	*p*
Mean number of injections	1.69	1.26	0.02	3.05	2.42	0.02
	Baseline	6 months after photocoagulation	*p*	12 months after photocoagulation		*p*
Mean BCVA (LogMAR)	0.47	0.41	0.08	0.39		0.11
Mean foveal thickness (µm)	403	336	<0.01	317		<0.01
Mean focal thickness (µm)	492	427	<0.01	425		<0.01

**Table 4 pharmaceutics-15-00308-t004:** Subgroup analysis of the primary and secondary outcomes according to the size of the CMA (BCVA—Best-Corrected Visual Acuity; CMA—Capillary Macroaneurysm).

CMA > 150 µm						
	6 months before photocoagulation	6 months after photocoagulation	*p*	12 months before photocoagulation	12 months after photocoagulation	*p*
Mean number of injections	2.30	1.40	<0.001	4.05	2.84	<0.01
	Baseline	6 months after photocoagulation	*p*	12 months after photocoagulation		*p*
Mean BCVA (logMAR)	0.36	0.34	0.97	0.33		0.16
Mean foveal thickness (µm)	373	317	<0.01	309		<0.01
Mean focal thickness (µm)	474	401	<0.01	402		<0.01
**CMA < 150 µm**						
	6 months before photocoagulation	6 months after photocoagulation	*p*	12 months before photocoagulation	12 months after photocoagulation	*p*
Mean number of injections	2.91	1.73	<0.01	5.15	3.13	<0.01
	Baseline	6 months after photocoagulation	*p*	12 months after photocoagulation		*p*
Mean BCVA (logMAR)	0.26	0.26	0.95	0.25		0.60
Mean foveal thickness (µm)	319	306	0.34	280		0.002
Mean focal thickness (µm)	441	397	<0.01	395		<0.01

**Table 5 pharmaceutics-15-00308-t005:** Subgroup analysis of the primary and secondary outcomes according to the cause of macular edema (BCVA—Best-Corrected Visual Acuity, RVO—Retinal Vein Occlusion).

Diabetes						
	6 months before photocoagulation	6 months after photocoagulation	*p*	12 months before photocoagulation	12 months after photocoagulation	*p*
Mean number of injections	2.52	1.53	<0.01	4.37	3.15	<0.01
	Baseline	6 months after photocoagulation	*p*	12 months after photocoagulation		*p*
Mean BCVA (logMAR)	0.32	0.30	0.73	0.28		0.06
Mean foveal thickness (µm)	360	319	<0.01	299		<0.01
Mean focal thickness (µm)	476	415	<0.01	413		<0.01
**RVO**						
	6 months before photocoagulation	6 months after photocoagulation	*p*	12 months before photocoagulation	12 months after photocoagulation	*p*
Mean number of injections	2.51	1.48	<0.01	4.58	2.59	<0.01
	Baseline	6 months after photocoagulation	*p*	12 months after photocoagulation		*p*
Mean BCVA (logMAR)	0.33	0.33	0.70	0.33		0.68
Mean foveal thickness (µm)	342	302	0.02	298		<0.01
Mean focal thickness (µm)	436	371	0.02	373		<0.01

## Data Availability

The data presented in this study are available on request from the corresponding author. The data are not publicly available due to privacy.

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
