# Peer review of "Focal Photocoagulation as an Adjunctive Therapy to Reduce the Burden of Intravitreal Injections in Macula Edema Patients, the LyoMAC2 Study"

_pharmaceutics, 2023, doi:10.3390/pharmaceutics15020308_

Round 1

Reviewer 1 Report

Dear authors,

I find your paper very interesting but the scientific presentation of the study could be improved.

The true nature of the paper which is a non-randomized retrospective study has to stand out more clearly.

The multiple bias: retrospective, non-randomized, monocentric, few doctors doing the laser and taking the clinical decision, have to be addressed and discussed.

In the discussion, authors mention that it is an open label study. This term is restricted only for prospective study. Using such a word could confuse the readers.

Also, authors argue that the strength of the study is its design but it is a retrospective study, there is no design. Only the proper analysis of the data could be valued.

Although intuitive, the focal edema thickness has not been defined.

The Table 3 is not clear enough and has to be improved for the reader.

Moreover, factual errors such as vascular growth epithelial factor instead of endothelial are in the manuscript.

Author Response

We thank Reviewer 1 for his careful review. Please see the attachment.

Reviewer 2 Report

The paper is well written and the results are well presented and make intuitive sense. However, the study design makes it difficult to draw any conclusions from the data. The main problem is that the cases come from various centers and doctors who appear to be acting independently, governed by general guidelines. The treatment decisions appear to be to some degree subjective, and several different drugs with various degrees of efficacy and duration of effect were employed, also reflecting surgeon preference. Both the injection frequency and drug choice may have been influenced by the fact that laser was done, rather than only based on the effect of the laser; or may have simply reflected the continuing long-term treatment for the ME which tends to require fewer injections with time, rather than showing any effect of the laser treatment. Further, the change in injection frequency before / after PC was ~ 1 injection / time interval on average. While statistically significant, this also represents the smallest possible incremental injection rate change / eye.

No CMAs within 750um of the fovea were treated. However, it is difficult to imagine such lesions farther from the fovea being a significant problem. Apparently about 2/3 of the CMA were between 1400-3100um from the fovea. It is unclear how such lesions so far from the fovea could significantly influence outcomes based on central foveal thickness and VA, presumably the priorities determining the need for injection. The fact that the location of the CMA based on tertiles stratified by distance from the fovea did not seem to influence the outcome / injection frequency calls into question whether the CMA were actually important at all, and whether treating the CMA actually altered the course of the disease.  This could be addressed with non-lasered controls, but none are available. 

It is appealing to think that PC of the CMAs was helpful. However, I don't think it is possible to say so based on this current study.  The doctors may have been influenced to give fewer injections knowing that laser had been done; or the passage of time may have simply reduced the need for injections in any case.

Author Response

We thank Reviewer 2 for his careful review. Please see the attachment.

Reviewer 3 Report

The authors presented a the LyoMAC2 study, using Focal photocoagulation as an adjunctive therapy to reduce the burden of intravitreal injections in macula edema patients.

The research appeals to me but I have following major concerns

Literature is not explored well nor I can see justification of the proposed approach.

Main contributions not highlighted.

How authors claim their approach is better than those reported in state of art.

Dataset description missing

Conclusion is too short

Mostly ref. are too old.

I suggest authors to enhance model & include on research framework

Include one section for literature review/background too

Author Response

We thank Reviewer 3 for his careful review. Please see the attachment.

Round 2

Reviewer 2 Report

I appreciate the authors responses to my concerns. However, my concerns reflect structural failings of the study. In their responses to my comments the authors frequently use terms like "seems" "intuitive" and express "regret" that the results were not in some cases what they expected. Such comments illustrate my concern that no conclusions can be supported by the study due it is design and that they reflect a prior bias favoring focal laser treatment. The advantage from laser is indeed intuitive. But it has not been established by the study. 

Reviewer 3 Report

Corrections well addressed & so accepted in present form.